# Evaluating the Hospital Standardized Home-Transition Ratios for Cerebral Infarction in Japan: A Retrospective Observational Study from 2016 through 2020

**DOI:** 10.3390/healthcare10081530

**Published:** 2022-08-13

**Authors:** Ryo Onishi, Yosuke Hatakeyama, Kanako Seto, Koki Hirata, Kunichika Matsumoto, Tomonori Hasegawa

**Affiliations:** Department of Social Medicine, Toho University School of Medicine, 5-21-16, Omori-Nishi, Tokyo 143-8540, Japan

**Keywords:** cerebral infarction, hospital to home transition, quality indicator, administrative data, Japan

## Abstract

Discharge to home is considered appropriate as a treatment goal for diseases that often leave disabilities such as cerebral infarction. Previous studies showed differences in risk-adjusted in-hospital mortality and readmission rates; however, studies assessing the rate of hospital-to-home transition are limited. We developed and calculated the hospital standardized home-transition ratio (HSHR) using Japanese administrative claims data from 2016–2020 to measure the quality of in-hospital care for cerebral infarction. Overall, 24,529 inpatients at 35 hospitals were included. All variables used in the analyses were associated with transition to another hospital or facility for inpatients, and evaluation of the HSHR model showed good predictive ability with c-statistics (area under curve, 0.73 standard deviation; 95% confidence interval, 0.72–0.73). All HSHRs of each consecutive year were significantly correlated. HSHRs for cerebral infarction can be calculated using Japanese administrative claims data. It was found that there is a need for support for low HSHR hospitals because hospitals with high/low HSHR were likely to produce the same results in the following year. HSHRs can be used as a new quality indicator of in-hospital care and may contribute to assessing and improving the quality of care.

## 1. Introduction

In the past few decades, many countries have been facing similar challenges related to healthcare, such as a rapidly aging population, increase in the prevalence of chronic illnesses, rising healthcare expenditures, and maldistribution of medical resources. In Japan, increasing focus has been developed to manage medical resources to reduce burden and increase the quality of life of patients. Additionally, the shift from in-hospital care to home care has been promoted as a healthcare policy. The hospital-to-home transition rate is partly related to the medical fee system in Japan [1,2,3,4,5]. Cerebral vascular disease (CVD) is one of the typical diseases common among the elderly. CVD is the leading cause of death and imposes a burden of patients worldwide. In Japan, CVD is one of the leading causes of morbidity and hospitalization, with a large proportion of patients having cerebral infarction. In 2017, there were 31.2 thousand deaths due to cerebral infarction among a total of 90.4 thousand inpatients and 60.2 thousand outpatients [6]. Inpatients with cerebral infarction have a high probability of discharge with continued attention to recurrence and have a long length of hospital stay; therefore, in-hospital care, including discharge management and support, is important. Previous studies have shown that optimal care during hospitalization might reduce the mortality and the readmission rate and the length of hospital stay [7,8,9,10,11]. Moreover, these studies have shown differences in risk-adjusted in-hospital mortality and readmission rates.

Conceptually, adjustment of patient characteristics and risks is needed to compare and evaluate hospitals with different capacities and functions. The hospital standardized mortality ratio (HSMR) is a risk-adjusted practical model for evaluating in-hospital care [12]. The HSMR is an important measure to improve patient safety and quality of care in hospitals. In the HSMR calculation, risk of in-hospital mortality is adjusted for factors such as patient age, sex, severity, comorbidities, and admission status. It then compares between the actual and risk-adjusted number of in-hospital deaths. The HSMR contributes to identifying areas for improvement to help reduce them. Actually, HSMR has been used in many developed countries and regions, such as Australia, Canada, France, Hong Kong, Japan, Singapore, Sweden, the UK, and the USA [13,14,15,16]. Additionally, adjustment methodologies for readmission risk have been used in the Hospital Readmission Reduction Program and previous studies [17]. We believe that the risk-adjusted hospital-to-home transition rate is one of the new medical quality indicators for evaluating acute care hospitals in an aging society. Administrative claims data submitted from hospitals have been broadly used for research to analyze the quality of care. For Japanese acute hospitals, the Diagnostic Procedure Combination/Per Diem Payment System (DPC/PDPS) is the main medical service reimbursement system. In DPC/PDPS, acute care hospitals are required to submit the data of day-by-day medical services such as medication, laboratory tests, and procedures as well as patients’ personal attributes such as comorbidities, activity of daily life, and prognosis. With the DPC/PDPS data being submitted regularly by hospitals, a big database is automatically built, and is now used for public reporting, revising tariffs, and research [3,18,19,20,21]. Administrative claims data have been successfully used for designing healthcare policies and disease management [22,23]. Thus far, there are limited studies assessing the rate of hospital-to-home transition, which needs focus.

This study aimed to develop a method for calculating the risk-adjusted home-transition ratio (hospital standardized home-transition ratio (HSHR)) for cerebral infarction using DPC data and clarify characteristics of them in Japan. 

## 2. Materials and Methods

### 2.1. Data and Calculation Model

We used anonymized DPC/PDPS data from the Medi-Target benchmarking project managed by the All Japan Hospital Association (AJHA), which is one of the largest nationwide hospital associations in Japan. The benchmarking project uses clinical indicators for improving the quality of hospital care based on DPC/PDPS data. Participation in the project was optional, and there were 60 participating hospitals in 2020 [24]. In this study, we used the data of the patients whose primary diagnosis was cerebral infarction, who were admitted directly from their home, and were discharged alive in fiscal years 2016–2020. Only hospitals that provided data for all years from 2016 to 2020 were included in the analysis. 

We constructed two analysis models for analyses to calculate the hospital-level risk-adjusted home-transition ratio, namely, the single-year HSHR model, which used each year’s data, and the five-year HSHR model, which used the 2016–2020 data. 

All inpatients with a primary diagnosis of cerebral infarction at admission were identified using the DPC/PDPS. The 10th revision of the International Statistical Classification of Diseases and Related Health Problems (ICD-10) code I63 was used to determine the diagnosis. The HSHR was defined as the ratio of the actual number of home-transitions aggregated at the hospital level to the expected number of home-transitions aggregated at the hospital level multiplied by 100. Home was defined as the patient’s own home and elderly housing with long-term care services according to the guideline for the reimbursement of the public medical insurance in Japan. The observed number of home-transitions is the sum of the actual number of home-transitions, and the expected number of home-transitions is based on the sum of the probabilities of home-transitions. Coefficients derived from logistic regression models are used to calculate the probability of home-transition. An HSHR above 100 indicates that the home-transition ratio is higher than the overall average.
$$\mathrm{HSHR}=\left(\frac{\sum^{\;}\mathrm{Observed}\;\mathrm{number}\;\mathrm{of}\;\mathrm{home}-\mathrm{transitions}}{\sum^{\;}\mathrm{Expected}\;\mathrm{number}\;\mathrm{of}\;\mathrm{home}-\mathrm{transitions}}\right)\times 100$$

A multivariable logistic regression analysis was performed to predict the chance of home-transition for each patient with patient-level factors. Logistic regression analyses were performed for the risk adjustment of patient characteristics and to calculate the intercept of the covariates. Coefficients derived from the logistic regression analysis were used to calculate the probability of home-transition. The sum of the predicted probabilities of home-transitions (range: 0–1) provided the total expected number of home-transitions in each hospital. The ratio of the actual number of home-transitions to the expected number of home-transitions provided information on the standardized ratio for that hospital of interest.

This study was based on a secondary analysis of DPC/PDPS data. Owing to the anonymous nature of the data, no Institutional Review Board approval was required for this kind of study in Japan [25]. This study was adjudicated as not applicable for ethical review by the Ethics Committee of Toho University School of Medicine (No. A19053). 

### 2.2. Statical Analyses

We used logistic regression analysis to adjust for the risk of aggravation of a patient’s state during hospitalization. Specifically, we calculated hospital performance for each outcome during the study period as the patient-level case-mix variables described below to control for the patient’s state at admission and during hospitalization. 

Using the DPC database from the AJHA, we assessed the hospital-level risk-adjusted performance of the home-transition rate. In the logistic regression analysis, the data included six control variables related to state of cerebral infarction. Patient data included information on age, sex, use of ambulance, surgery, pre-stroke Rankin scale (PRS), Carlson comorbidity index (CCI), and discharge destination. As for risk-adjusted variables, age, sex, urgency of admission, and CCI were used for risk adjustment in previous studies [8,26,27]. The CCI is a weighted score based on the number and type of diagnoses reported in the hospital medical information [28,29,30]. The CCI was based on secondary ICD-10 diagnosis codes. In the DPC/PDPS, the top four serious comorbidities were reported. In this study, the CCI was calculated based on Quan’s modification and classified into four categories: CCI score 0, 1–2, 3–4, and 5 or over (5+) [31].

In this study, we adopted surgery and PRS as additional control variables for risk-adjustment of severity. The PRS was categorized into four levels: 1, 2, 3, and 4. As patient characteristics differed between hospitals, it was necessary to adjust for the patient’s risk of severity [32]. Patient characteristics of difference in the discharge destination were compared using chi-square tests for categorical variables and *t*-tests for continuous variables. 

For evaluating the predictive accuracy of logistic models, the c-statistic was used, which is derived by calculating the proportion of concordant pairs. A c-statistic value of 0.5 suggests that the model is no better than random chance in predicting death, and a value of 1.0 indicates perfect discrimination. For evaluating the relationships between the HSHRs for each year, Spearman’s correlation coefficient was used. 

All *p* values were two-sided at an alpha level of 0.05. All statistical analyses were performed using the Statistical Package for the Social Sciences (SPSS), version 27.0.0.

## 3. Results

### 3.1. Study Sample

From April 2016 to March 2020, a total of 24,529 inpatients were included at 35 hospitals. The demographic characteristics of patients are presented in Table 1. Overall, 65.5% of the patients were discharged home. The mean (±standard deviation (SD)) age was 73.0 years in the discharge-to-home group and 77.9 ± 10.9 years in the discharge-to-another facility group. The percentage of patients who used an ambulance at admission was 50.8% (40.8% in the discharge-to-home group and 69.7% in the discharge-to-another facility group), and there was no planned admission. In this study, 8.7% patients underwent surgery during hospitalization (4.9% in the discharge-to-home group and 15.8% in the discharge-to-another facility group). The demographic characteristics of the patients were similar for each year (Table 2).

### 3.2. Logistic Regression Analysis for Adjusting Patient Risk

Chi-square tests and *t*-test show that all risk-adjusted variables were associated with differences in the discharge destination (discharge-to-home or discharge-to-another facility) in Table 1 and Table 2. The results of the coefficients and significance of the variables are shown in Table 3. All variables show a significant relationship with difference in the discharge destination. The results of the coefficients and significance for single-year HSHR mode are shown in Table 4. Although the results of CCI were a little different, the results were broadly similar to the five-year HSHR model. The results of c-statistics showed predictive abilities of 0.73 (95% confidence interval (CI), 0.72–0.73) in all period analyses, and 0.73 (95%CI, 0.72–0.75), 0.72 (95%CI, 0.71–0.74), 0.72 (95%CI, 0.71–0.74), 0.73 (95%CI, 0.71–0.74), and 0.73 (95%CI, 0.71–0.74) in 2016, 2017, 2018, 2019, and 2020, respectively, in the analyses of each year.

### 3.3. HSHRs

HSHRs varied widely across the hospitals (Figure 1). Table 5 shows the mean (±SD) of the HSHRs, which was stable in each year, ranging from 102.28 ± 23.66 in 2020 to 103.69 ± 19.96 in 2017. In the five-year model, the mean HSHR (±SD) was 103.18 ± 19.02 and the HSHR in each hospital ranged from 43.38 to 136.27 (Figure 2), and the percentage of hospitals with HSHR higher than 100 was 57.1%. The correlation analyses reveal a significant positive relationship between the changes in HSHRs in each consecutive year (Table 6). The high/low HSHR hospitals have a trend to continue their HSHR.

Positive correlation coefficient means that hospitals with lower/higher HSHRs are likely to have similar results in the following year.

## 4. Discussion

This study showed that HSHRs for inpatients with cerebral infarction can be calculated using DPC/PDPS data. The DPC/PDPS is a standard reimbursement system for acute care hospitals, and almost all acute care hospitals submit all medical service data electronically, therefore, these results can be widely applied. The calculation method developed in this study could be used to assess the quality of inpatient care, especially hospital-to-home transition for cerebral infarction. Discharge to home with discharge management support is considered appropriate as a treatment goal for diseases that often leave disabilities such as cerebral infarction. These results showed that the HSHRs varied considerably among hospitals with comparable case-mixes. To the best of our knowledge, this is the first large-scale study to calculate HSHRs for cerebral infarction in Japan.

After adjustments of a patient’s risks, some hospitals were found to have a lower home-transition ratio. Regarding the patients’ age, 82.6% of the admitted patients were 65 years and older. Aging is known to contribute to aggravation of cerebral infarction. The *t*-test and chi-square test results for the relationship between discharge destination and control variables were significant. In this study, confounding variables could not be adjusted; however, the variables used in regression analysis were considered appropriate.

For the risk-adjusted method, we used the logistic regression analysis because it has been used in previous studies and could also be conducted to evaluate the quality of care in each hospital. For risk-adjusting severity before the onset of cerebral infarction, we used PRS, which was utilized in the DPC/PDPS database. In this study, we found that hospitals with high/low HSHR had the trend of producing similar results in the following year. It was suggested that HSHR for cerebral infarction is a stable quality indicator.

As for future studies, it will be necessary to consider the characteristics and discharge support systems applied by good HSHR hospitals so that poor HSHR hospitals can benefit from improving the home-transition rate [33,34]. To improve the quality of hospital care, it is important for hospitals to check the quality of their care management, including discharge management. Participating the benchmarking allows evaluation and comparison of the care levels [35]. The quality indicator of in-hospital care, especially increasing home-transition rate, is important for management from the perspective of maintaining a patient’s life [5]. In Japan, the shift from long hospitalization to home care has been promoted, which may be expanded in the future. Therefore, HSHR based on DPC data could be considered a useful indicator for hospital managers and policy makers in healthcare. 

The strengths of our study included large sample size, developing the risk-adjusted methods for new quality indicator, and the results of c-statistics that showed predictive ability. However, this study has some limitations. First, the hospitals assessed in this study might not be representative of all hospitals in Japan because they voluntarily participated in the benchmarking project. We consider that our future analysis will focus on the HSHR using all DPC/PDPS data in Japan with the methodology developed in this study. Second, other risk factors of cerebral infarction, such as blood pressure level, medical history, social economic status, and hospital function were not considered in this study [36,37]. Third, we could not examine patients whose condition worsened after discharge in this study, but previous studies indicated that care after discharge is important [38,39,40].

## 5. Conclusions

In this study, it is possible to calculate the HSHRs for cerebral infarction using administrative claims data from the DPC/PDPS database. The HSHRs as a new quality indicator showed variation among hospitals with comparable case-mixes. In this study period, since the hospitals with high/low HSHR were likely to produce similar results in the following year, the necessity for healthcare policy intervention in low HSHR hospitals was identified. In a super-aged society, discharge to home is considered appropriate as an in-hospital treatment goal for diseases that often leave disabilities. The HSHR for cerebral infarction might contribute to the development of more useful quality indicators for hospitals to improve their quality of care.

## Figures and Tables

**Figure 1 healthcare-10-01530-f001:**
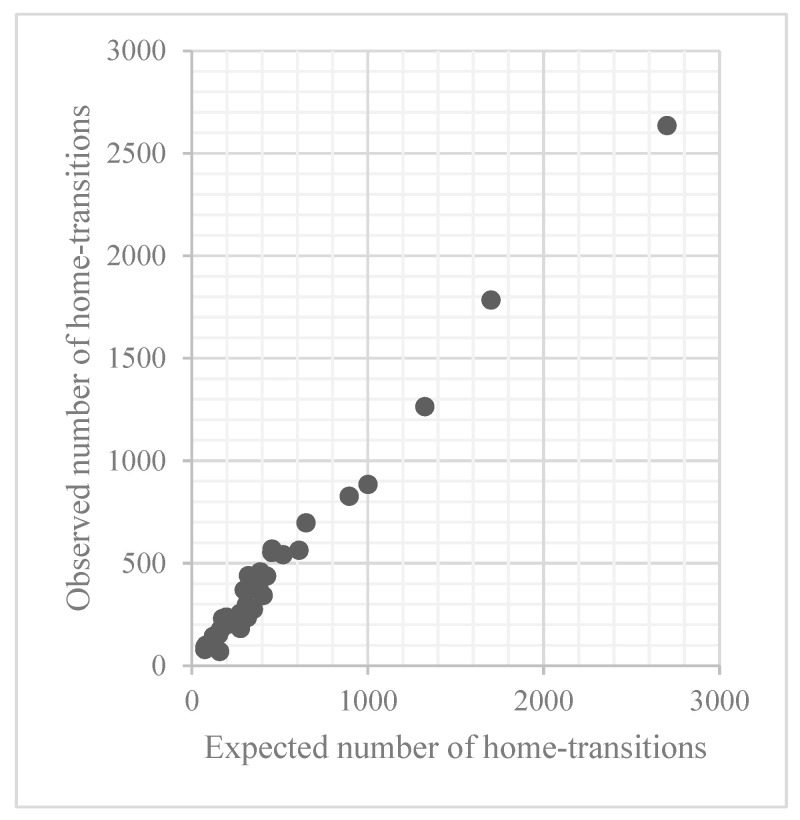
Variation in HSHR. HSHR = hospital standardized home-transition ratio.

**Figure 2 healthcare-10-01530-f002:**
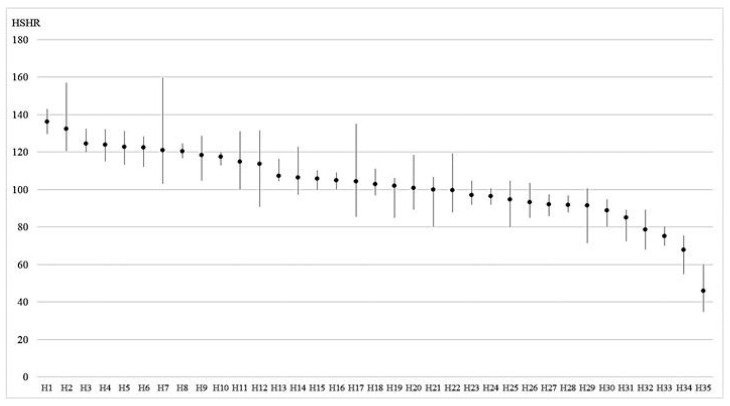
Mean and width * of HSHR in each hospital. H = hospital. HSHR = hospital standardized home-transition ratio. *Width means the range between the minimum and maximum HSHR.

**Table 1 healthcare-10-01530-t001:** Demographic characteristics of patients in five-year HSHR model.

Characteristic		Total (*n* = 24,529)	Discharge to Home (*n* = 16,073)	Discharge to Another Facility (*n* = 8456)	*p* Values *
Age	mean ± SD	74.6 ± 12.0	73.0 ± 12.1	77.9 ± 10.9	<0.001
Male sex	n (%)	14,726 (60.0)	9988 (62.1)	4738 (56.0)	<0.001
Use of ambulance	n (%)	12,456 (50.8)	6564 (40.8)	5892 (69.7)	<0.001
Surgery	n (%)	2133 (8.7)	795 (4.9)	1338 (15.8)	<0.001
PRS	n (%)				<0.001
PRS 1		17,553 (71.6)	12,228 (76.1)	5325 (63.0)	
PRS 2		2972 (12.1)	1896 (11.8)	1076 (12.7)	
PRS 3		1956 (8.0)	1037 (6.5)	919 (10.9)	
PRS 4		1563 (6.4)	731 (4.5)	832 (9.8)	
PRS 5		485 (2.0)	181 (1.1)	304 (3.6)	
CCI	n (%)				<0.001
CCI score 0		17,670 (72.0)	11,994 (74.6)	5676 (67.1)	
CCI score 1–2		5963 (24.3)	3581 (22.3)	2382 (28.2)	
CCI score 3–4		773 (3.2)	437 (2.7)	336 (4.0)	
CCI score 5+		123 (0.5)	61 (0.4)	62 (0.7)	

HSHR = hospital standardized home-transition ratio. n = number of inpatients. PRS = pre-stroke Rankin scale. CCI = Charlson comorbidity index. *p* values = two-sided significance. * Patient characteristics were compared using chi-square tests for categorical variables and *t*-test for continuous variable (Age).

**Table 2 healthcare-10-01530-t002:** Demographic characteristics of patients in single-year HSHR models.

		2016		2017		2018		2019		2020	
Characteristic		Discharge to Home (*n* = 3324)	Discharge to Another Facility (*n* = 1718)	Discharge to Home (*n* = 3369)	Discharge to Another Facility (*n* = 1680)	Discharge to Home (*n* = 3257)	Discharge to Another Facility (*n* = 1687)	Discharge to Home (*n* = 3322)	Discharge to Another Facility (*n* = 1758)	Discharge to Home (*n* = 2801)	Discharge to Another Facility (*n* = 1613)
Age	mean ± SD	72.3 ± 12.0	77.2 ± 11.0	72.5 ± 12.1	77.6 ± 11.0	73.5 ± 11.9	78.3 ± 10.9	73.4 ± 12.4	78.2 ± 10.8	73.2 ± 12.4	78.2 ± 11.1
Male sex	n (%)	2085 (62.7)	990 (57.6)	2102 (62.4)	937 (55.8)	2027 (62.2)	936 (55.5)	2029 (61.1)	969 (55.1)	1745 (62.3)	906 (56.2)
Use of ambulance	n (%)	1340 (40.3)	1203 (70.0)	1396 (41.4)	1142 (68.0)	1353 (41.5)	1171 (69.4)	1312 (39.5)	1216 (69.2)	1163 (41.5)	1160 (71.9)
Surgery	n (%)	138 (4.2)	257 (15.0)	150 (4.5)	228 (13.6)	159 (4.9)	275 (16.3)	169 (5.1)	299 (17.0)	179 (6.4)	279 (17.3)
PRS	n (%)										
PRS 1		2498 (75.2)	1029 (59.9)	2623 (77.9)	1082 (64.4)	2523 (77.5)	1107 (65.6)	2494 (75.1)	1092 (62.1)	2090 (74.6)	1015 (62.9)
PRS 2		412 (12.4)	221 (12.9)	372 (11.0)	200 (11.9)	352 (10.8)	205 (12.2)	387 (11.6)	248 (14.1)	373 (13.3)	202 (12.5)
PRS 3		213 (6.4)	226 (13.2)	202 (6.0)	162 (9.6)	196 (6.0)	154 (9.1)	241 (7.3)	192 (10.9)	185 (6.6)	185 (11.5)
PRS 4		165 (5.0)	171 (10.0)	131 (3.9)	165 (9.8)	152 (4.7)	166 (9.8)	159 (4.8)	168 (9.6)	124 (4.4)	162 (10.0)
PRS 5		36 (1.1)	71 (4.1)	41 (1.2)	71 (4.2)	34 (1.0)	55 (3.3)	41 (1.2)	58 (3.3)	29 (1.0)	49 (3.0)
CCI	n (%)										
CCI score 0		2414 (72.6)	1125 (65.5)	2497 (74.1)	1117 (66.5)	2450 (75.2)	1142 (67.7)	2504 (75.4)	1191 (67.7)	2129 (76.0)	1101 (68.3)
CCI score 1–2		784 (23.6)	514 (29.9)	761 (22.6)	478 (28.5)	716 (22.0)	463 (27.4)	726 (21.9)	483 (27.5)	594 (21.2)	444 (27.5)
CCI score 3–4		111 (3.3)	67 (3.9)	92 (2.7)	67 (4.0)	84 (2.6)	65 (3.9)	81 (2.4)	80 (4.6)	69 (2.5)	57 (3.5)
CCI score 5+		15 (0.5)	12 (0.7)	19 (0.6)	18 (1.1)	7 (0.2)	17 (1.0)	11 (0.3)	4 (0.2)	9 (0.3)	11 (0.7)

HSHR = hospital standardized home-transition ratio. n = number of inpatients. PRS = pre-stroke Rankin scale. CCI = Charlson comorbidity index.

**Table 3 healthcare-10-01530-t003:** Variables in the logistic regression analysis in five-year HSHR model.

	OR (95% CI)	*p* Values
Age	0.97 (0.97–0.97)	<0.001
Male sex	1.08 (1.02–1.15)	0.011
Use of ambulance	0.34 (0.32–0.36)	<0.001
Surgery	0.32 (0.29–0.35)	<0.001
PRS 1 (reference)		
PRS 2	0.83 (0.76–0.91)	<0.001
PRS 3	0.61 (0.55–0.67)	<0.001
PRS 4	0.49 (0.43–0.54)	<0.001
PRS 5	0.42 (0.35–0.52)	<0.001
CCI score 0 (reference)		
CCI score 1–2	0.86 (0.81–0.92)	<0.001
CCI score 3–4	0.85 (0.73–1.00)	0.044
CCI score 5+	0.55 (0.37–0.80)	0.002

HSHR = hospital standardized home-transition ratio. PRS = pre-stroke Rankin scale. CCI = Carlson comorbidity index. OR = odds ratio. *p* values = two-sided significance.

**Table 4 healthcare-10-01530-t004:** Variables in the logistic regression analysis in single-year HSHR models.

	2016		2017		2018		2019		2020	
	OR (95% CI)	*p* Values	OR (95% CI)	*p* Values	OR (95% CI)	*p* Values	OR (95% CI)	*p* Values	OR (95% CI)	*p* Values
Age	0.97 (0.97–0.98)	<0.001	0.97 (0.96–0.98)	<0.001	0.97 (0.96–0.98)	<0.001	0.97 (0.97–0.98)	<0.001	0.97 (0.97–0.98)	<0.001
Male sex	1.03 (0.90–1.17)	0.703	1.09 (0.95–1.24)	0.212	1.10 (0.97–1.26)	0.145	1.16 (0.98–1.27)	0.098	1.07 (0.93–1.23)	0.323
Use of ambulance	0.32 (0.29–0.37)	<0.001	0.36 (0.32–0.41)	<0.001	0.35 (0.31–0.40)	<0.001	0.33 (0.29–0.38)	<0.001	0.31 (0.27–0.35)	<0.001
Surgery	0.29 (0.23–0.36)	<0.001	0.33 (0.26–0.41)	<0.001	0.28 (0.23–0.35)	<0.001	0.31 (0.25–0.38)	<0.001	0.39 (0.31–0.48)	<0.001
PRS 1 (reference)										
PRS 2	0.83 (0.69–1.01)	0.062	0.84 (0.69–1.02)	0.080	0.80 (0.66–0.98)	0.032	0.73 (0.61–0.89)	0.001	0.97 (0.79–1.19)	0.775
PRS 3	0.46 (0.37–0.58)	<0.001	0.61 (0.48–0.77)	<0.001	0.67 (0.53–0.85)	0.001	0.72 (0.58–0.90)	0.003	0.62 (0.49–0.79)	<0.001
PRS 4	0.53 (0.42–0.68)	<0.001	0.41 (0.31–0.53)	<0.001	0.49 (0.38–0.64)	<0.001	0.52 (0.40–0.66)	<0.001	0.47 (0.36–0.32)	<0.001
PRS 5	0.36 (0.23–0.55)	<0.001	0.37 (0.25–0.57)	<0.001	0.44 (0.28–0.69)	<0.001	0.47 (0.31–0.73)	0.001	0.53 (0.32–0.87)	0.011
CCI score 0 (reference)										
CCI score 1–2	0.86 (0.74–0.99)	0.036	0.87 (0.76–1.01)	0.074	0.90 (0.78–1.04)	0.163	0.87 (0.75–1.00)	0.057	0.80 (0.69–0.94)	0.006
CCI score 3–4	1.10 (0.78–1.54)	0.596	0.89 (0.63–1.26)	0.494	0.91 (0.64–1.31)	0.620	0.62 (0.44–0.87)	0.006	0.79 (0.54–1.17)	0.237
CCI score 5+	0.87 (0.38–2.01)	0.745	0.55 (0.27–1.12)	0.100	0.19 (0.08–0.49)	0.001	1.75 (0.50–6.10)	0.378	0.48 (0.18–1.26)	0.136

HSHR = hospital standardized home-transition ratio. PRS = pre-stroke Rankin scale. CCI = Charlson comorbidity index. OR = odds ratio. *p* values = two-sided significance.

**Table 5 healthcare-10-01530-t005:** Mean and SD of HSHRs.

Year	Mean	SD
2016	103.10	23.02
2017	103.69	19.96
2018	102.32	17.87
2019	102.28	18.90
2020	102.76	23.66
2016–2020	103.18	19.02

SD = standard deviation. HSHR = hospital standardized home-transition ratio.

**Table 6 healthcare-10-01530-t006:** Correlation between the HSHRs in each consecutive year.

Period	r	*p* Values
2016–2017	0.81	<0.001
2017–2018	0.89	<0.001
2018–2019	0.80	<0.001
2019–2020	0.74	<0.001

HSHR = hospital standardized home-transition ratio. r = correlation coefficient (Spearman’s non-parametric correlation). *p* values = two-sided significance.

## Data Availability

In this research, limited databases provided by the All Japan Hospital Association were used, and the raw data are restricted by a Japanese regulation, Act on the Protection of Personal Information, and cannot be shared publicly. External researchers can contact the Ethics Committee of Toho University regarding the use of data but the committee only accepts applications in Japanese (med.rinri@ext.toho-u.ac.jp, +81-3-3762-4151). External researchers may contact the research team directly in English for assistance with their application: health@med.toho-u.ac.jp, ryo.onishi@med.toho-u.ac.jp.

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
