# Peer review of "Evaluating the Hospital Standardized Home-Transition Ratios for Cerebral Infarction in Japan: A Retrospective Observational Study from 2016 through 2020"

_healthcare, 2022, doi:10.3390/healthcare10081530_

Round 1

Reviewer 1 Report

This manuscript reported a secondary analysis of data on standardized home-transition ratio in Japan. The authors did not follow the Instruction for Authors, especially reference style.

Points to note are:

Results

1.     Line 167: ‘2020’ should be ‘2019’

2.     Table S2 has not been mentioned in the main text.

Patents

3.     This section is not necessary and should be deleted.

References

4.     All the references are in the wrong format. There are too many errors to be listed. Some examples are given below.

5.     For documents co-authored by a large number of persons (more than 10 authors), you can either cite all authors, or cite the first ten authors add a semicolon and add ‘et al.’ at the end.

6.     Abbreviated journal name should be used

7.     Year of publication should be in bold

8.     Comma should be used before and after the volume number

9.     pp. should not be used

10.  Incorrect style of referencing a website

11.  Reference 11: The number 11 was duplicated. There should be no issue number. Article number is missing.

12.  Reference 28: Delete the gap within the URL

Author Response

Dear Reviewer 1,

Thank you for your comments and suggestions.

We revised our manuscript according to your comments.

Reviewer 2 Report

Peer review report for the manuscript Evaluating the hospital standardized home-transition ratios for cerebral infarction in Japan: A retrospective observational study from 2016 through 2020.

The aim of this manuscript is to develop a method for calculating the risk-adjusted home-transition ratio (hospital standardized home-transition ratio [HSHR]) for cerebral infarction using DPC data and clarify characteristics of them in Japan.

The manuscript is well written and organized. The authors clearly explained the complexity of their study approach, methodology, data analysis, and findings step by step. The study was conducted rigorously. The sample size was large enough to produce robust results. I have a few comments.

In introduction authors should add more information about HSMR. We do not know anything about risk factors.

The material and method prepared very clearly. Authors added all important information to describe methodological aspect of the research.

Results: the Table S1 should be add to the main text (section 3.1) not in supplementary and the Table 1 should be deleted. The same situation with Table 2.

Authors should add the figures, tables and schemes in the place where they cite them, not in other section (so the section 3.4 should be deleted).

Authors used to much self-citation.

I not sure that Special Issue: Patient Care Assessment is adequate place for this publication.  

Thanks for the opportunity to review the manuscript.

Author Response

Dear Reviewer,

Thank you for your comments and suggestions.

We revised our manuscript according to your comments.

This manuscript is a resubmission of an earlier submission. The following is a list of the peer review reports and author responses from that submission.